# Early-Stage Ovarian Malignancy Score versus Risk of Malignancy Indices: Accuracy and Clinical Utility for Preoperative Diagnosis of Women with Adnexal Masses

**DOI:** 10.3390/medicina56120702

**Published:** 2020-12-16

**Authors:** Phichayut Phinyo, Jayanton Patumanond, Panprapha Saenrungmuaeng, Watcharin Chirdchim, Tanyong Pipanmekaporn, Apichat Tantraworasin, Theera Tongsong, Charuwan Tantipalakorn

**Affiliations:** 1Department of Family Medicine, Faculty of Medicine, Chiang Mai University, Chiang Mai 50200, Thailand; phichayutphinyo@gmail.com; 2Center for Clinical Epidemiology and Clinical Statistics, Faculty of Medicine, Chiang Mai University, Chiang Mai 50200, Thailand; jpatumanond@gmail.com; 3Department of Obstetrics and Gynecology, Faculty of Medicine, Mahasarakham University, Maha Sarakham 44150, Thailand; Pps2pearkabpom@gmail.com; 4Department of Obstetrics and Gynecology, Phrapokklao Hospital, Chanthaburi 22000, Thailand; watcharin.ch@cpird.in.th; 5Department of Anesthesiology, Faculty of Medicine, Chiang Mai University, Chiang Mai 50200, Thailand; tanyong24@gmail.com; 6Department of Surgery, Faculty of Medicine, Chiang Mai University, Chiang Mai 50200, Thailand; ohm_med@hotmail.com; 7Department of Obstetrics and Gynecology, Faculty of Medicine, Chiang Mai University, Chiang Mai 50200, Thailand; theera.t@cmu.ac.th

**Keywords:** ovarian cancer, adnexal mass, clinical utility, decision curve analysis, net benefit

## Abstract

*Background and objectives:* To compare the diagnostic accuracy and clinical utility of the Early-stage Ovarian Malignancy (EOM) score with the Risk of Malignancy Index (RMI) in the presurgical assessment of women presenting with adnexal masses. *Materials and Methods:* A secondary analysis was carried out in a retrospective cohort of women who presented with an adnexal mass and were scheduled for surgery at Phrapokklao Hospital between September 2013 and December 2017. The clinical characteristics, ultrasonographic features of the masses, and preoperative CA-125 levels were recorded. The EOM and the RMI score were calculated and compared in terms of accuracy and clinical utility. Decision curve analysis (DCA), which examined the net benefit (NB) of applying the EOM and the RMI in practice at a range of threshold probabilities, was presented. *Results:* In this study, data from 270 patients were analyzed. Fifty-four (20.0%) women in the sample had early-stage ovarian cancer. All four RMI versions demonstrated a lower sensitivity for the detection of patients with early-stage ovarian cancer compared to an EOM score ≥ 15. An EOM ≥ 15 resulted in a higher proportion of net true positive or NB than all versions of the RMIs from a threshold probability of 5% to 30%. *Conclusions:* It also showed a higher capability to reduce the number of inappropriate referrals than the RMIs at a threshold probability between 5% and 30%. The EOM score showed higher diagnostic sensitivity and has the potential to be clinically more useful than the RMIs to triage women who present with adnexal masses for referral to oncologic gynecologists. Further external validation is required to support our findings.

## 1. Introduction

An accurate preoperative diagnosis of women who present with adnexal masses is essential for general gynecologists to arrive at the appropriate management decisions [1]. Generally, all women with ovarian tumors should be referred for subjective assessment by experienced sonographers, which is currently accepted as the most accurate and effective decision-making tool for women with ovarian tumors [2]. However, expert examiners are not widely available, especially in resource-limited settings. Thus, the clinical burden of a preoperative diagnosis of adnexal masses usually falls in the hand of general gynecologists. Women with a high probability of malignant ovarian tumors should then be referred to gynecologic oncologists for specialized oncological managements, as it is evident that this approach leads to a more favorable survival outcome for the patients [3,4]. In contrast, women with benign ovarian tumors should not be referred to avoid excessive stress for patients, creation of long waiting times, unnecessary healthcare costs, and, at the worst, overly radical surgical interventions [5]. Expectant and conservative surgical management by general gynecologists may be more appropriate in this domain of patients to reduce morbidity and preserve fertility [6].

Over the last three decades, several diagnostic algorithms and multimodal tests have been developed and recommended for use in practice to assist the differentiation of ovarian tumors. These include the International Ovarian Tumor Analysis (IOTA) systems (e.g., IOTA Simple Rules and IOTA logistic regression model (LR2)), the Risk of Ovarian Malignancy Algorithm (ROMA), and the Risk of Malignancy Index (RMI) [7,8]. Each diagnostic tool uses different predictive parameters for ovarian malignancy. The IOTA systems rely mainly on specific ultrasonographic features, whereas the ROMA focuses on serum biomarkers. The classic RMI combines multiple parameters, including menopausal status, simple ultrasonographic patterns, and serum CA-125, a widely recognized ovarian malignancy marker. A recent meta-analysis in 2014 concluded that the IOTA Simple Rules and the IOTA LR2 were the diagnostic rules with the highest accuracy level [9]. Even though the IOTA Simple Rules carried high diagnostic performance, up to about 10–20% of the examinations were inconclusive and required subsequent specialist consultations [10,11]. Due to this disadvantage of the IOTA Simple Rules, the RMI is still the most widely used and the most frequently validated decision tool for presurgical differentiation of ovarian pathology because of its simplicity and conclusive results [1,9].

The main disadvantage of the RMI is that it is heavily influenced by serum CA-125 levels, which is not a sensitive marker in the case of type I epithelial tumors (i.e., low grade serous, endometrioid, mucinous, and clear cell carcinoma) and early-stage ovarian cancer (i.e., stage I-IIA and IIIA1 according to the Fédération Internationale de Gynécologie et d’Obstétrique (FIGO) staging) [12,13,14]. Besides, high levels of serum CA-125 are not solely specific to ovarian cancer as it can be elevated in other benign conditions, for example, endometriosis [15]. In Thailand and some other Asian countries, there is a high prevalence of endometriosis and type I epithelial tumors; therefore, the application of RMI might not be appropriate and may lead to patient misclassification [16,17]. A validation study of all four RMI versions was done in Thailand in 2013 [18]. It was found that the diagnostic accuracy and discriminative ability of the RMI were not as high as previously claimed [19,20,21,22].

In 2019, the Early-stage Ovarian Malignancy (EOM) score was developed from a study in a cohort of women with benign ovarian tumors and early-stage ovarian cancer [23]. The EOM score was calculated based on the same parameters as the RMI. It demonstrated excellent discriminative ability with an area under the receiver operating characteristic curve of 0.88 and with good calibration. More evidence is needed to confirm the diagnostic ability and clinical usefulness of the EOM score as a proper triage tool for patient referral to specialized oncology care. This study aimed to evaluate the diagnostic accuracy and the clinical utility of the recently developed EOM score against all four versions of the RMI scoring in the presurgical assessment of women presenting with adnexal masses.

## 2. Materials and Methods

A secondary analysis was conducted using a retrospective cohort of women that presented with an adnexal mass at Phrapokklao Hospital from September 2013 to December 2017. Phrapokklao Hospital is a university-affiliated tertiary care hospital located in Chanthaburi Province located in the eastern seaboard of Thailand. The Research in Human Ethical Committee of Chanthaburi Province (CTIREC 046) and the Institutional Ethics Committee of the Faculty of Medicine, Chiang Mai University, approved the study protocol (Research ID: 7254 No. 179/2020). Informed consent was waived as data were collected retrospectively and anonymously. Patient data were kept confidential and in compliance with the Declaration of Helsinki.

The study included all women with an adnexal mass who presented consecutively at the hospital and were scheduled for surgery during the study period. Patients were eligible regardless of the type of ovarian operation. Key data collection comprised the preoperative clinical characteristics, the ultrasonographic features of the mass, and the tumor marker (CA-125), which were recorded during routine case management. The routine ultrasonographic assessment was generally performed by non-expert sonographers (including residents in training and general gynecologists). All ultrasonographic images were reevaluated and verified by a single gynecologic oncologist (WC). Ultrasound examination had been carried out by connecting a 3.5–5 MHz transabdominal or a 5–7.5 MHz transvaginal transducer to a sonoaceX7 (Samsung, South Korea). If multiple masses were present in a single patient, aspects of the mass with the most complex ultrasonographic features or the mass with the largest maximal diameter were collected.

The histopathologic reports of the resected masses were reviewed. Only patients with the diagnoses of benign ovarian tumors and early-stage ovarian cancer were included. In this study, early-stage ovarian cancer was defined as FIGO stage I, II, and IIIA1 (microscopic) [24,25]. Data from patients with advanced-stage ovarian cancer, recurrent ovarian cancer, metastatic cancer to the ovary, patients with a history of neoadjuvant chemotherapy, and patients with incomplete data were excluded. Patients with a pathological diagnosis of borderline ovarian tumors were included in the early-stage ovarian cancer group.

In this study, we planned to evaluate the diagnostic accuracy and the clinical utility of 5 diagnostic risk models for preoperative discrimination of benign and malignant adnexal tumors: EOM, RMI-I, RMI-II, RMI-III, and RMI-IV. Table 1 illustrates the components and score rating of each risk model.

The Early-stage Ovarian Malignancy score or the EOM score was recently developed from the dataset used in the primary data analysis prior to this study [23]. The EOM score is a combination of five simple predictors: menopausal status, tumor size, the presence of a solid component, the presence of ascites, and serum CA-125. Each component of the EOM score was rated with a different scoring regime, which was based on the logit coefficients of the prediction model. The total EOM score, ranging from 0 to 51, was derived from the summation of the score from each component. In the case of the EOM score, two cutoff points were proposed to categorize patients into low, moderate, and high-risk groups. The score cutoff point at ≥15 was intended to be used by general gynecologists to make a referral decision, whereas the score cutoff point at ≥30 was for oncologic specialists to prioritize patients for surgery or further investigation.

The Risk of Malignancy Index (RMI) is the classic diagnostic tool for preoperative diagnosis of ovarian cancer, which was developed in 1990 [19]. The RMI incorporates three main components, which are menopausal status, the ultrasound score, and the serum CA-125 level. The total score is the multiplicative product of all three components. The cutoff point for suggesting a high risk of a malignant tumor was originally set at an RMI ≥200 [26]. This cutoff point was generally used as the criteria for referring patients to a specialist. Currently, there are four versions of the RMI: RMI I [19], RMI II [20], RMI III [21], and RMI IV [22]. The differences in each version of the RMI give different score ratings for each component in RMI I to RMI III. In the fourth version of the RMI, tumor size is included in the score calculation, and a different cutoff point of >450 had been proposed.

All statistical analyses were performed using Stata version 16 (StataCorp, College Station, TX, USA). Frequency and percentage were used to describe the categorical data. The mean and standard deviation or median (interquartile range) were used to describe the continuous data, as appropriate. The comparison of categorical data was made using the exact probability test. The comparison of continuous data was made using an independent *t*-test or Mann–Whitney U test according to the data distribution. Two-tailed *p*-values < 0.05 were considered as statistically significant.

Diagnostic accuracy indices, sensitivity, specificity, positive predictive values, negative predictive values, and positive likelihood ratios were calculated for each risk model. The 95% confidence intervals were estimated using the Clopper Pearson’s binomial exact method. The comparison of sensitivity and specificity between the EOM score and each version of the RMI was carried out using an exact McNemar’s test. The area under the receiver operating characteristic curve (AuROC) was used to represent the discriminative ability of the risk models. We compared the equality of the AuROC of each RMI model against the EOM score and used Sidak’s adjustment for multiple comparisons.

To evaluate the clinical utility of the risk models, we employed the concept of decision curve analysis (DCA), which was introduced in 2006 by Vickers et al. [27]. DCA focuses on the net benefit (NB) gained from the application of risk models in practice. NB indicates the proportion of the net true positives, which remain after considering the presence of false positives. In a clinical context, the benefits of true positives and the harm of false positives are generally on different scales. Weighting factors must be used to convert false positives to the same scale as the true positives [28]. In this study, we focused on the appropriate referrals of patients to oncologic specialists. Thus, the weighting factor corresponds to the odds of the chosen threshold probability (T) to refer the patients to specialists (odds = T/1 − T). For instance, a threshold probability of 10% for patient referral implies that up to nine false positives are acceptable per one true positive (odds or benefit-to-harm ratio = 1/9). In summary, the NB is calculated as follows:$$\mathrm{Net}\;\mathrm{Befit}=\frac{(\mathrm{Number}\;\mathrm{of}\;\mathrm{True}\;\mathrm{positives}-(\mathrm{Number}\;\mathrm{of}\;\mathrm{False}\;\mathrm{positives}\;\times \;\left(T÷\left(1-T\right)\right)}{\left(\mathrm{Total}\;\mathrm{sample}\;\mathrm{size}\right)}$$
where T is the threshold probability.

However, in actual practice, the choice of threshold probability varies largely depending on the discretion of each physician, as the reasonable threshold involves holistic incorporation of all potential disease outcomes. It is essential to evaluate the NB over a range of reasonable threshold probabilities within the clinical context of interest, as there is no single acceptable threshold probability [29]. In this study, we estimated the NB of all the risk models (at pre-specified cutoff points) from the risk threshold probability range of 5% to 50%, which was based on a report from a recent study on the clinical utility of the International Ovarian Tumor Analysis (IOTA) risk models [5]. We focused primarily on two important threshold probabilities, 10% and 25%, as these points were generally used as cutoff points for malignancy in the IOTA logistic regression model [30].

The decision curves were plotted to visualize the trend in NB for each model across the range of threshold probabilities. The NBs of two default strategies were also included in the graph for comparison: referring all patients or referring none. The NB of any risk model should be higher than the NB of the default strategies; otherwise, the model would be considered clinically harmful in its application. As the statistical significance and confidence intervals are not important in classical decision theory [31], we did not present the confidence intervals and *p*-values for the decision curve analysis.

We also presented an alternative expression of NB by calculating the net proportion of true negatives, which can be interpreted as the number of inappropriate referrals avoided per 100 patients. The equation used is as follows:$$\mathrm{Net}\;\mathrm{proportion}\;\mathrm{of}\;\mathrm{True}\;\mathrm{negatives}=\;\frac{\mathrm{NB}-{\mathrm{NB}}_{\mathrm{ReferAll}}}{\mathrm{Odds}\left(T\right)}$$
where *NB* is the net benefit; and *T* is the threshold probability.

## 3. Results

During the study period, there were 640 women with an adnexal mass scheduled for surgery and who were assessed for eligibility. Of this number, 370 women were excluded (Figure 1). Two hundred and seventy patients were therefore included in the analysis. The mean age of the patients was 44.5 ± 13.0 years. Only 25.6% of the patients (*n* = 69) were of postmenopausal age. One-third of the patients (*n* = 90, 33.3%) were nulliparous. Of the 270 patients included in the analysis, 216 (80.0%) were diagnosed with benign ovarian tumors, and 54 (20.0%) with early-stage ovarian cancer. In this dataset, there were eight patients with borderline ovarian tumors. All eight patients were included in the early-stage ovarian cancer group. The histopathological classification of all ovarian tumors included in the study is presented in Table 2. The comparison of the clinical characteristics, ultrasonographic features, and tumor marker levels between patients with ovarian cancer and patients with benign ovarian tumors are shown in Table 3.

The sensitivity of the EOM score at the cutoff point of ≥15 was significantly higher than all versions of the RMI (*p* < 0.001), while the specificity was significantly lower (*p* < 0.001 for RMI I, RMI III, and RMI IV, and *p* = 0.002 for RMI II). In contrast, the EOM score at the higher cutoff point of ≥30 showed a significantly higher specificity than that of all RMI versions (*p* < 0.001). However, at this EOM cutoff, the sensitivity was significantly lower than all RMI versions (*p* < 0.001). The other diagnostic indices (positive predictive values, negative predictive values, and positive likelihood ratios) are illustrated in Table 4. With regard to the discriminative ability via the AuROC, the EOM score was significantly more superior than RMI-I (0.88 vs. 0.73, *p* < 0.001), RMI-II (0.88 vs. 0.76, *p* < 0.001), RMI-III (0.88 vs. 0.75, *p* < 0.001), and RMI-IV (0.88 vs. 0.78, *p* = 0.002) (Figure 2).

Regarding clinical utility, the EOM score at the cutoff point of ≥15 demonstrated a higher proportion of net true positive or NB than all versions of the RMIs at the pre-specified cutoffs from the threshold probabilities of 5% to 30% (Table 5). At the threshold probabilities of 10% and 25%, a NB of 0.158 and 0.098 by the EOM score means that there are 15.8 and 9.8 net-detected early-stage ovarian cancers per 100 patients without inappropriate referrals, respectively. At the threshold probability beyond 30%, the NB of the EOM score decreased. Using all versions of the RMI for guiding referral of patients with adnexal masses was clinically harmful at a threshold probability lower than 10%, as an approach of referral of all patients to oncologic specialists shows a higher NB. If the threshold probability for referral was between 10% and 15%, the RMIs were not considered harmful, as the NB of all the RMIs were higher than both the default strategies (refer all and refer none). However, at this risk threshold, the NB of the EOM score was higher than that of the RMIs (Figure 3a).

The EOM score also showed a higher capability to reduce inappropriate referrals than the RMIs from the threshold probabilities of 5% to 30% (Table 5) (Figure 3b). In comparison to an approach of referring all patients to oncologists, using the EOM score could reduce the number of inappropriate referrals by up to 43 per 100 patients at the risk threshold of 10% and 48 per 100 patients at the threshold probability of 20%. In contrast, at the threshold probability of 10%, only the RMI-II at the cutoff of 200 could reduce the number of inappropriate referrals by about 3 per 100 patients, whereas the other RMIs increased the number of inappropriate referrals. At a risk threshold of 25%, the EOM score reduced inappropriate referrals by between 5 and 8 (per 100 patients) more than when the RMI was used.

## 4. Discussion

In this study, all four RMI versions at their pre-specified cutoff points demonstrated a lower sensitivity for detecting patients with early-stage ovarian cancer when compared to the EOM score ≥15. Although the specificity of the EOM score at this cutoff was significantly lower than the RMI, its negative predictive value was the highest and may be more effective as a triage tool for patient referrals. An EOM score ≥15 also demonstrated clinical utility over a wide range of threshold probabilities, from as low as 5% up to 30%. The RMI had less clinical utility and was considered harmful at a low threshold probability but was beneficial at a higher threshold probability (>30%). Our findings suggest that, in some specific situations, the EOM score at the cutoff point of ≥15 should be used instead of the RMI to increase the detection of ovarian cancer cases while minimizing the number of inappropriate patient referrals.

In a recent meta-analysis, the pooled estimates of the sensitivity and specificity of the RMI I were 72.0% (95% CI: 67.0, 76.0%) and 92.0% (95% CI: 89.0, 93.0%), respectively [9]. The pooled sensitivity and specificity of the other RMI versions were comparable to the first version. The original publication of the RMI in 1990 reported the sensitivity and specificity of the RMI as 85.4% and 95.9% [19], respectively, which was significantly higher than the number reported in the meta-analysis and also from our findings. Out of all patients with malignant diseases included in the original study, around two-thirds (28/42, 66.7%) of the patients were at advanced stages of ovarian cancer or metastatic disease. As the RMI scoring is largely dependent on the serum CA-125 levels, a biomarker that is more specific to an advanced stage or type II ovarian cancer [14], the diagnostic performance of the RMI is undoubtedly disturbed in studies that include a lower proportion of these patients [18,32,33,34,35]. Our study only included patients with early-stage ovarian cancer, which probably explains the lower sensitivity of all the RMIs.

The EOM score at the cutoff point of ≥15 outperformed all four RMI versions in terms of diagnostic sensitivity in the detection of patients with early-stage ovarian cancer. The ability to discriminate between benign and early malignant conditions of the EOM score was also significantly higher than in all the RMIs. The derivation of the EOM score from a cohort of patients with benign ovarian tumors and early-stage ovarian cancer improves the ability to distinguish both conditions. Moreover, the EOM scoring system appropriately adjusts the influence of the serum CA-125 level into the model by categorizing this classic marker into three categories, assigning a weighted score to each. In our prior work, we proposed two cutoff points, at ≥15 and ≥30, for the implementation of the EOM score in practice [23]. The finding from this study adds weight to the existing evidence that the lower cutoff (≥15) is suitable for triage purposes, while the higher cutoff (≥30) might be more appropriate for diagnosis and giving guidance for the prioritization of patients for surgery.

Even though the EOM score exhibited good discrimination and calibration properties, this does not justify and establish its usefulness in making appropriate decisions for the referral of patients with a high probability of ovarian cancer to gynecologic oncologists [28]. In this study, we examined the clinical utility of the EOM score and the RMI in supporting decision-making by applying a novel decision-analytic approach, or DCA [27]. One of the most troublesome challenges to this approach lies within the definition of a reasonable threshold probability of patient referral [36], as the threshold generally depends on multiple factors, such as the judgement of the physician, patient preferences, local referral guidelines, and availability of healthcare resources [5]. As early detection of ovarian cancer would result in early treatment, appropriate staging, and survival improvements [6,37], it is more important to detect one case of ovarian cancer than to avoid referrals to oncologists (i.e., sensitivity is more important than specificity [1]). In our setting, patients with benign ovarian tumors who were misclassified and referred to specialists, or false-positive cases, usually undergo further non-invasive investigation before being scheduled for invasive surgery. In contrast, patients with ovarian cancer who otherwise would not be referred, or false-negative cases, might be subject to more unfavorable outcomes in this context, such as inadequate staging of the cancer [38,39] or suboptimal surgical management [40]. 

The NB of the EOM score at the cutoff point of ≥15 was higher than the approach of referring all patients, not referring any patients, and all the RMI versions from a threshold probability of 5% up to 30%. The application of the EOM score would also result in a lower number of inappropriate referrals. This was due to the higher sensitivity of the EOM score compared to the RMI. At a lower threshold probability, the detection of one ovarian cancer case weighs substantially more than one inappropriate referral. Thus, when false-positive cases are acceptable, a diagnostic rule with higher sensitivity would gain a higher NB. However, when false-positive cases are unacceptable, a prediction model with high specificity is warranted. In practice, a decision to refer patients only when the threshold probability was higher than 50% was not sufficiently reasonable, as it implies that referring false-positive cases is more harmful than not referring any patients [5]. We focused on the clinical utility of the EOM score and the RMI at two clinically relevant thresholds, 10% and 25%. We concluded that the EOM score ≥15 was clinically more useful in informing general gynecologists regarding the preoperative assessment of patients with adnexal masses than the RMI. The RMI would be useful only when the threshold probability was higher than 30%.

There were some limitations to be addressed. First, this study was a secondary analysis of a retrospective cohort study used to develop the EOM score. Thus, the accuracy and clinical utility of the EOM score might be overestimated. Further external validation studies on both aspects should be performed and are currently in process to support the findings in this study. Second, only patients with adnexal masses who were scheduled for surgery were included in this study. Patients with benign conditions who were conservatively managed were excluded, which might account for a large number of patients. However, this enables us to base all the diagnoses on the more reliable reference standard, histopathologic examinations. Third, the routine ultrasonographic evaluations were performed by non-expert sonographers, which limits the generalizability of our results to the setting where ultrasound examination was performed by an experienced sonographer. Nonetheless, by using data from a routine examination, our results could be considered as pragmatic regarding the accuracy and utility of both models. Fourth, only early-stage ovarian cancer cases were included. At this point, the transferability of our results could not be extended to patients with advanced-stage cancer or metastatic diseases. Fifth, our study included patients with FIGO stage IIIA as early-stage ovarian cancer following a previous study [25], which might not be conventional and could overestimate the discriminative ability of both the EOM score and the RMIs. However, as the overestimation occurs equally in both groups, differential bias during a comparative validation is unlikely. Finally, we did not examine the effect modification of menopausal status on the diagnostic accuracy and clinical utility as our study size in terms of menopausal women was limited and might not provide adequate statistical power. Therefore, subgroup effects might be present and should not be overlooked until further evidence is available.

## 5. Conclusions

In conclusion, the EOM score provides high diagnostic sensitivity, making it more suitable to use as a triage tool for referring women with an adnexal mass to specialized oncologic care than all versions of the RMI. It also demonstrates potential clinical utility by increasing the net detection of early-stage ovarian cancer cases without raising the number of inappropriate referrals. The results are promising but an external validation study would be required to confirm the diagnostic performance and clinical utility of the EOM score before being endorsed for clinical use.

## Figures and Tables

**Figure 1 medicina-56-00702-f001:**
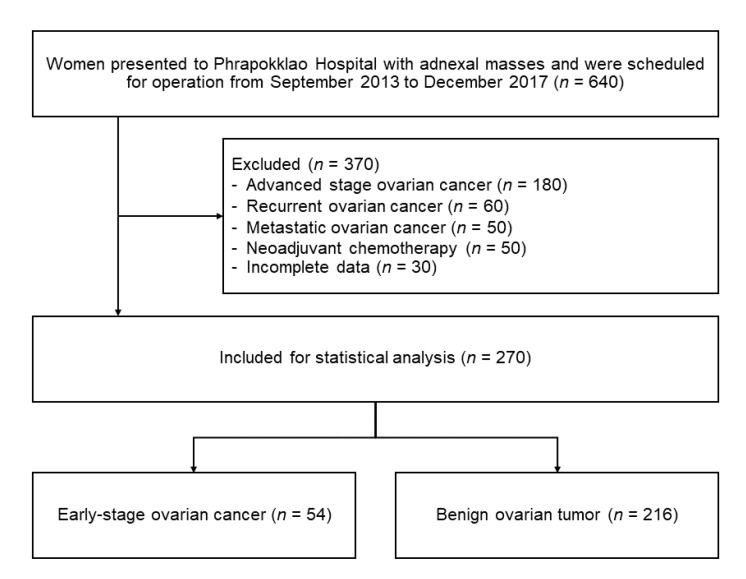
Study flow diagram.

**Figure 2 medicina-56-00702-f002:**
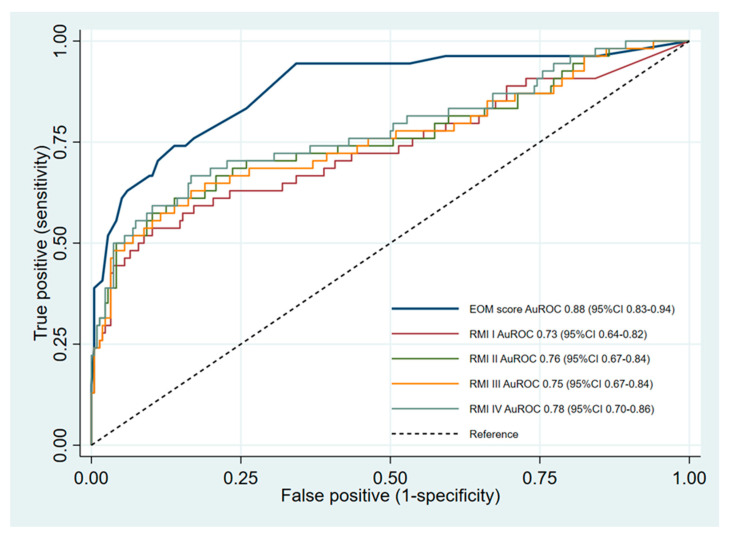
Comparative validation of the discriminative ability between the EOM scores and the RMIs via the receiver operating characteristic curve. Abbreviations: AuROC, area under the receiver operating characteristic curve; EOM, Early-stage Ovarian Malignancy; RMI, Risk of Malignancy Indices.

**Figure 3 medicina-56-00702-f003:**
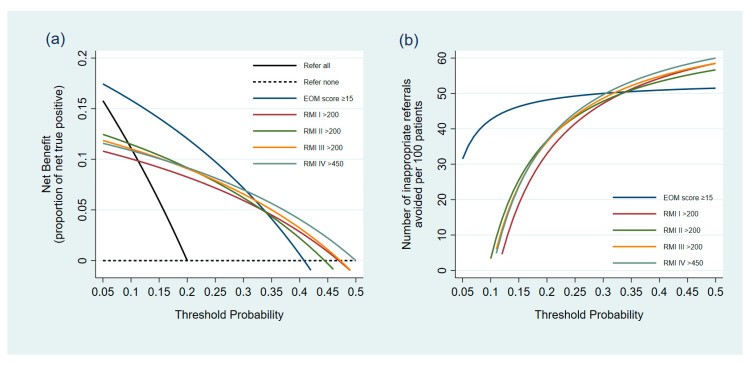
Comparative validation of the clinical utility between the EOM scores and Risk of Malignancy Indices via decision curves. (**a**) Decision curve plotting the net benefit against threshold probability. (**b**) Decision curve plotting a decrease in inappropriate referrals against threshold probability.

**Table 1 medicina-56-00702-t001:** Components of the Early-stage Ovarian Malignancy (EOM) score and Risk of Malignancy Index (RMI) I–IV.

Components	EOM	RMI ^†^			
		I	II	III	IV
**Menopausal Status**					
Pre-menopause	0	1	1	1	1
Post-menopause	7	3	4	3	4
**Ultrasonographic Features**					
None	0	0	1	1	1
Any 1 feature		1	1	1	1
Presence of multilocularity					
Presence of solid component	3				
Bilateral lesions					
Presence of ascites	13				
Presence of intra-abdominal metastases					
≥2 Features		3	4	3	4
**Tumor Size (cm)**					
<7					1
≥7					2
<9	0				
9–12	10				
>12	16				
**Serum CA125 (IU/L)**					
<30	0				
30–200	1				
>200	12				

Abbreviations: EOM, Early-stage Ovarian Malignancy score; RMI, Risk of Malignancy Index. ^†^ For the calculation of the RMIs, the total score is the multiplicative product of the summation of the menopausal score, ultrasound score, tumor size score (for RMI IV), and serum CA-125.

**Table 2 medicina-56-00702-t002:** Histopathological classification of the ovarian tumors.

	*n* (%)
**Malignant Tumors** (*n* = 46)	
Endometrioid carcinoma	11 (20.4)
Serous carcinoma	10 (18.5)
Mucinous carcinoma	6 (11.1)
Adenocarcinoma	6 (11.1)
Clear cell carcinoma	5 (9.3)
Granulosa cell tumor	3 (5.6)
Other rare tumors	5 (9.3)
**Borderline Tumors** (*n* = 8)	
Borderline mucinous tumor	7 (12.9)
Borderline serous tumor	1 (1.8)
Borderline endometrioid tumor	0 (0)
**Benign Tumors** (*n* = 216)	
Endometriotic cyst	77 (35.7)
Dermoid cyst	45 (20.8)
Mucinous cystadenoma	41 (19.0)
Serous cystadenoma	26 (12.0)
Follicular cyst	11 (5.1)
Corpus luteal cyst	10 (4.6)
Others rare tumors	6 (2.8)

**Table 3 medicina-56-00702-t003:** Characteristics of the study patients.

Variables	Early Stage Ovarian Cancer (*n* = 54)		Benign Ovarian Tumor (*n* = 216)		*p*-Value
	*n*	(%)	*n*	(%)	
**Clinical Characteristics**					
Age (year) *	48.7	±15.4	43.5	±12.1	0.008
Nulliparity	21	(38.9)	69	(31.9)	0.337
Post-menopause	28	(51.9)	41	(19.0)	<0.001
**Ultrasonographic Features**					
Maximum tumor diameter (cm) *	16.4	±6.7	10.1	±5.1	<0.001
Multilocularity	38	(70.4)	145	(67.1)	0.745
Solid component	33	(61.1)	57	(26.4)	<0.001
Bilateral lesions	5	(9.3)	43	(19.9)	0.075
Ascites	11	(20.4)	2	(0.9)	<0.001
Intra-abdominal metastases	2	(3.7)	1	(0.5)	0.103
**Tumor Marker**					
Serum CA125 (IU/L) **	102.8	25.0, 314.0	30.8	14.8, 81.1	<0.001

Abbreviations: SD, standard deviation; IQR, interquartile range. * Mean ± SD, ** Median (IQR).

**Table 4 medicina-56-00702-t004:** Comparative diagnostic performance of the Early-stage Ovarian Malignancy (EOM) score and Risk of Malignancy Indices (RMI).

Scores	Cutoff	Early Stage Ovarian Cancer (*n* = 54)		Benign Ovarian Tumor (*n* = 216)		Sensitivity (%) (95% CI)	Specificity (%) (95% CI)	PPV (%) (95% CI)	NPV (%) (95% CI)	LR+ (95% CI)
		*n*	(row%)	*n*	(row%)					
EOM	≥15	51	(40.8)	74	(59.2)	94.4 (84.6–98.8)	65.7 (59.0–72.0)	40.8 (32.1–49.9)	97.9 (94.1–99.6)	2.76 (1.68–4.50)
	<15	3	(2.1)	142	(97.9)					
	≥30	21	(95.5)	1	(4.6)	38.9 (25.9–53.1)	99.5 (97.4–100.0)	95.5 (77.2–99.9)	86.7 (81.8–90.7)	84.0 (12.73, 3488.41)
	<30	33	(13.3)	215	(86.7)					
RMI I	≥200	31	(47.0)	35	(53.0)	57.4 (43.2–70.8)	83.8 (78.2–88.4)	47.0 (34.6–59.7)	88.7 (83.6–92.7)	3.54 (1.92–6.49)
	<200	23	(11.3)	181	(88.7)					
RMI II	≥200	36	(44.4)	45	(55.6)	66.7 (52.5–78.9)	79.2 (73.1–84.4)	44.4 (33.4–55.9)	90.5 (85.4–94.3)	3.20 (1.81–5.61)
	<200	18	(9.5)	171	(90.5)					
RMI III	≥200	34	(47.2)	38	(52.8)	63.0 (48.7–75.7)	82.4 (76.7–87.2)	47.2 (35.3–59.3)	89.9 (84.8–93.7)	3.58 (1.98–6.43)
	<200	20	(10.1)	178	(89.9)					
RMI IV	≥450	33	(50.0)	33	(50.0)	61.1 (46.9–74.1)	84.7 (79.2–89.2)	50.0 (37.4–62.6)	89.7 (84.7–93.5)	4.00 (2.17–7.32)
	<450	21	(10.3)	183	(89.7)					

Abbreviations: PPV, positive predictive value; NPV, negative predictive value; LR+, positive likelihood ratios; CI, confidence interval; EOM score, Early-stage Ovarian Malignancy score; RMI, Risk of Malignancy Index.

**Table 5 medicina-56-00702-t005:** Decision curve analysis.

Threshold Probability	Net Benefit (Proportion of Net True Positive)						Reduced Numbers of Inappropriate Referrals per 100 Patients				
	Refer all	EOM	RMI I	RMI II	RMI III	RMI IV	EOM	RMI I	RMI II	RMI III	RMI IV
5%	0.157	0.174	0.108	0.125	0.119	0.116	31.482	−94.815	−63.333	−74.815	−80.000
10%	0.111	0.158	0.100	0.115	0.110	0.109	42.593	−9.630	3.333	−0.741	−2.222
15%	0.059	0.141	0.092	0.104	0.101	0.101	46.296	18.765	25.556	23.951	23.704
20%	0	0.120	0.082	0.092	0.091	0.092	48.148	32.963	36.667	36.296	36.667
25%	−0.067	0.098	0.072	0.078	0.079	0.081	49.259	41.481	43.333	43.704	44.445
30%	−0.143	0.071	0.059	0.062	0.066	0.070	50.000	47.160	47.778	48.642	49.630
35%	−0.231	0.041	0.045	0.044	0.050	0.056	50.529	51.217	50.952	52.169	53.333
40%	−0.333	0.006	0.028	0.022	0.032	0.041	50.926	54.259	53.333	54.815	56.111
45%	−0.455	−0.035	0.009	−0.003	0.011	0.022	51.235	56.626	55.185	56.872	58.272
50%	−0.600	−0.085	−0.015	−0.033	−0.015	0	51.481	58.519	56.667	58.519	60.000

Abbreviations: EOM, Early-stage Ovarian Malignancy score; RMI, Risk of Malignancy Index.

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
