# Peer review of "Early-Stage Ovarian Malignancy Score versus Risk of Malignancy Indices: Accuracy and Clinical Utility for Preoperative Diagnosis of Women with Adnexal Masses"

_medicina, 2020, doi:10.3390/medicina56120702_

Round 1

Reviewer 1 Report

The manuscript submitted for review concerns an interesting and important topic of the diagnosis of ovarian tumors, the detection of ovarian cancer at an early stage and supporting decisions about the place and method of treatment. Moreover, the authors describe a new tool they invented, which is better than the well-known RMI. Of course, it requires external validation, but a publication on this topic popularizing this tool is important.

My comments:

  1. The limitation of the work is its retrospective nature and that the early stages were tested on the basis of the database on which the model was created
  2. The language requires grammatical and stylistic correction in several places.
  3. I don't consider FIGO IIIa to be an early stage
  4. The authors should mention that referring a patient with a suspected ovarian tumor to a gynecologist-oncologist is for subjective ultrasound evaluation performed by an experienced ultrasound examiner, which is the most effective decision-making in women with ovarian tumors.
  5. Verses 216-223 require a different presentation

Author Response

Reviewers’ comment to the manuscript [medicina-1014313]
entitled “Early-stage Ovarian Malignancy Score versus Risk of Malignancy Indices: accuracy and clinical utility for preoperative diagnosis of women with adnexal masses”

Dear Editor and reviewers,

            We would like to thank you for the opportunity to revise our manuscript to be qualified for publication in Medicina. We have revised and modified some parts of our manuscript as addressed in response to reviewers’ comments. We hope that our responses and revisions would substantially improve the quality of our manuscript and would be qualified for publication in the journal. If there were any further questions or minor points to be addressed or elaborated, please let us know. We would be more than eager to make any further revision.

Reviewer#1: The manuscript submitted for review concerns an interesting and important topic of the diagnosis of ovarian tumors, the detection of ovarian cancer at an early stage and supporting decisions about the place and method of treatment. Moreover, the authors describe a new tool they invented, which is better than the well-known RMI. Of course, it requires external validation, but a publication on this topic popularizing this tool is important.

  • The limitation of the work is its retrospective nature and that the early stages were tested on the basis of the database on which the model was created
    • We have acknowledged this point within the limitation section of the discussion part in line 325-327.
  • The language requires grammatical and stylistic correction in several places.
    • We would like to thank you for all your comments on the manuscript. As for the English editing, our manuscript had already undergone academic English editing by a native English proofreader at our institution (editing certificate attached with revision).
    • However, regarding your recent suggestion we have made another thorough review of the manuscript for grammatical error and punctuation mistakes, and we have corrected some of the errors as highlighted in the manuscript.
    • We also made some changes to Figure 1 study flow diagram.
      • Change the study domain box to: Women presented at Phrapokklao Hospital with adnexal masses and were scheduled for surgery between September 2013 and December 2017
      • Included in statistical analysis
    • I don't consider FIGO IIIa to be an early stage
      • In the development dataset, patients with ovarian cancer FIGO stage IIIa were included as early-stage ovarian cancer followed the definition defined by one previous study [1]. The study revealed that the 5-year survival for FIGO stage IIIa patients was 78% and was superior to other advanced stage patients, which is in accordance with a previous report [2].
      • Unfortunately, as we did not have the complete data on patients’ complete surgical staging at this time, we could not perform a sensitivity analysis by excluding patient with stage IIIa out of the study sample. Thus, the definition of early-stage ovarian cancer in our study needs to follow our previous one. By doing this, we understand that the discriminative ability of both the EOM score and the RMIs could be overestimated. However, as the overestimation occurs in both groups equally, differential bias during a comparative validation is unlikely.
      • We have added an acknowledgment to this point in the limitation section (in line 338-342) as follows: Our study included patients with FIGO stage IIIA as early-stage ovarian cancer following a previous study [1], which might not be conventional and could overestimate the discriminative ability of both the EOM score and the RMIs. However, as the overestimation occurs in both groups equally, differential bias during a comparative validation is unlikely.
    • The authors should mention that referring a patient with a suspected ovarian tumor to a gynecologist-oncologist is for subjective ultrasound evaluation performed by an experienced ultrasound examiner, which is the most effective decision-making in women with ovarian tumors.
      • We have mentioned subjective assessment as for your suggestion within the first paragraph of the introduction part (in line 49-54) as follows: Generally, all women with ovarian tumors should be referred for subjective assessment by experienced sonographers, which is currently accepted as the most accurate and effective decision-making tool for women with ovarian tumors. However, expert examiners are not widely available, especially in resource-limiting settings. Thus, the clinical burden of preoperative diagnosis of adnexal masses usually falls in the hand of general gynecologists.
    • Verses 216-223 require a different presentation
      • We have rewritten the sentences as you suggested (in line 218-222) as follows: The sensitivity of the EOM score at the cutoff point of ≥15 was significantly higher than all versions of the RMI (p<0.001), while the specificity was significantly lower (p<0.001 for RMI I, RMI III, and RMI IV, p=0.002 for RMI II). In contrast, the EOM score at the higher cutoff point of ≥30 showed a significantly higher specificity than that of all RMI versions (p<0.001). However, at this EOM cutoff, the sensitivity was significantly lower than all RMI versions (p<0.001).

References

  1. Kleppe M, van der Aa MA, Van Gorp T, et al (2016) The impact of lymph node dissection and adjuvant chemotherapy on survival: A nationwide cohort study of patients with clinical early-stage ovarian cancer. Eur J Cancer 66:83–90. https://doi.org/10.1016/j.ejca.2016.07.015
  2. Onda T, Yoshikawa H, Yasugi T, et al (1998) Patients with ovarian carcinoma upstaged to stage III after systematic lymphadenctomy have similar survival to Stage I/II patients and superior survival to other Stage III patients. Cancer 83:1555–1560

Reviewer 2 Report

I appreciate this opportunity to review the paper written by Dr Phinyo et al entitled “Early-stage Ovarian Malignancy Score versus Risk of Malignancy Indices: accuracy and clinical utility for preoperative diagnosis of women with adnexal masses.”.

This study evaluated to compare the diagnostic accuracy and the clinical utility of the Early-stage Ovarian Malignancy (EOM) score with the Risk of Malignancy Index (RMI) in the presurgical assessment of women presenting with adnexal masses. The author was presented that the EOM and the RMI scoring were calculated and compared in terms of accuracy and clinical utility. Decision curve analysis (DCA), which examined the net benefit (NB) of applying the EOM and the RMI in practice at a range of threshold probability. The author suggested that The EOM score showed higher diagnostic sensitivity and might be clinically more useful than the RMIs to triage women who presented with adnexal masses for referral to oncologic gynecologists.

This is a "delicious" paper

Author Response

Reviewers’ comment to the manuscript [medicina-1014313]
entitled “Early-stage Ovarian Malignancy Score versus Risk of Malignancy Indices: accuracy and clinical utility for preoperative diagnosis of women with adnexal masses”

Dear Editor and reviewers,

            We would like to thank you for the opportunity to revise our manuscript to be qualified for publication in Medicina. We have revised and modified some parts of our manuscript as addressed in response to reviewers’ comments. We hope that our responses and revisions would substantially improve the quality of our manuscript and would be qualified for publication in the journal. If there were any further questions or minor points to be addressed or elaborated, please let us know. We would be more than eager to make any further revision.

Reviewer#2: I appreciate this opportunity to review the paper written by Dr Phinyo et al entitled “Early-stage Ovarian Malignancy Score versus Risk of Malignancy Indices: accuracy and clinical utility for preoperative diagnosis of women with adnexal masses.”.

This study evaluated to compare the diagnostic accuracy and the clinical utility of the Early-stage Ovarian Malignancy (EOM) score with the Risk of Malignancy Index (RMI) in the presurgical assessment of women presenting with adnexal masses. The author was presented that the EOM and the RMI scoring were calculated and compared in terms of accuracy and clinical utility. Decision curve analysis (DCA), which examined the net benefit (NB) of applying the EOM and the RMI in practice at a range of threshold probability. The author suggested that The EOM score showed higher diagnostic sensitivity and might be clinically more useful than the RMIs to triage women who presented with adnexal masses for referral to oncologic gynecologists.

This is a "delicious" paper

  • Extensive editing of English language and style required
    • We would like to thank you for all your comments on the manuscript. As for the English editing, our manuscript had already undergone academic English editing by a native English proofreader at our institution (editing certificate attached with revision).
    • However, regarding your recent suggestion we have made another thorough review of the manuscript for grammatical error and punctuation mistakes, and we have corrected some of the errors as highlighted in the manuscript.